# Evaluation of Tissue Tropism and Horizontal Transmission of a Duck Enteritis Virus Vectored Vaccine in One-Day-Old Chicken

**DOI:** 10.3390/v16111681

**Published:** 2024-10-29

**Authors:** Yassin Abdulrahim, Yingying You, Linggou Wang, Zhixiang Bi, Lihua Xie, Saisai Chen, Benedikt B. Kaufer, Armando Mario Damiani, Kehe Huang, Jichun Wang

**Affiliations:** 1College of Veterinary Medicine, Nanjing Agricultural University, Nanjing 210095, China; yassin1989@163.com (Y.A.); grace081606@163.com (L.W.); 2Institute of Veterinary Immunology and Engineering, Jiangsu Academy of Agricultural Sciences, Nanjing 210014, China; hiyouyy@163.com (Y.Y.); zhixiangbi@163.com (Z.B.); lhxie999@163.com (L.X.); 13912965070@163.com (S.C.); 3College of Veterinary Sciences, Nyala University, Nyala P.O. Box 155, South Darfur, Sudan; 4College of Veterinary Medicine Shandong, Agricultural University, Taian 271018, China; 5College of Animal Science and Technology, Guangxi University, Nanning 530004, China; 6Institute of Virology, Freie Universität Berlin, Robert von Ostertag-Straße 7-13, 14163 Berlin, Germany; b.kaufer@fu-berlin.de; 7Veterinary Centre for Resistance Research (TZR), Freie Universität Berlin, 14163 Berlin, Germany; 8Laboratorio de Bioquímica e Inmunidad, Facultad de Ciencias Médicas, Universidad Nacional de Cuyo, Mendoza 5502, Argentina; 9Instituto de Medicina y Biología Experimental de Cuyo, Consejo Nacional de Investigaciones Científicas y Técnicas (IMBECU-CONICET), Mendoza 5500, Argentina

**Keywords:** vector vaccines, DEV-H5, HVT-VP2, one-day-old chickens

## Abstract

Herpesvirus of turkey (HVT) recombinant vector vaccines are widely used in the poultry industry. However, due to limitations in loading multiple foreign antigens into a single HVT vector, other viral vectors are urgently needed. Since chickens lack maternal immunity to duck enteritis virus (DEV), vector vaccines using DEV as a backbone are currently under study. Even though a recently developed DEV vector vaccine expressing the influenza hemagglutinin H5 of highly pathogenic avian influenza (DEV-H5) induces highly detectable anti-HA antibodies, safety issues hamper further vaccine development. In this work, tissue affinity and horizontal transmission in 1-day-old chickens were systematically evaluated after DEV-H5 vector vaccine inoculation. Sixty percent of DEV-H5-inoculated chickens died between day 2 and day 7 post-inoculation. The displayed clinical signs consisted of lethargy, anorexia, and diarrhea, and virus was shed in feces. Gross and/or histological lesions were recorded in the kidney, heart, intestine, liver, lung, and spleen. Moreover, DEV-H5 replication in intestinal cells caused an increment in interferon-α expression, while occluding junction proteins and ZO-1 expression were significantly upregulated. As a control, birds inoculated with a commercial recombinant turkey herpesvirus expressing the VP2 protein of the infectious bursal disease virus (HVT-VP2) vector vaccine showed neither clinical signs nor mortality. Overall, while the HVT-VP2 vaccine demonstrated complete safety in 1-day-old chickens, our potential DEV-H5 vaccine requires further attenuation for consideration as a vector vaccine candidate in chickens.

## 1. Introduction

In recent years, the control of many poultry diseases relies on the use of recombinant DNA technology. Today, there are multiple commercially available recombinant vaccines that have the advantages of preventing multiple diseases simultaneously and simplifying the vaccination schedule [1,2,3].

Turkey herpesvirus (HVT) is one of the most common vaccine vectors that have been utilized to control Marek disease and several other avian diseases, including Newcastle disease (ND), avian influenza (AI), infectious bursal disease (IBD), and infectious laryngotracheitis (ILT) by encoding heterologous antigen proteins as dual vaccines [2,4,5]. Duck enteritis virus (DEV), an alpha-herpesvirus, poses a significant threat to waterfowl, particularly ducks, geese, and swans, with mortality rates reaching as high as 100% in infected ducks [6]. Its large genome and specific host range have prompted research into its potential as a vector for recombinant multivalent vaccines [7,8]. This innovative approach aims to influence DEV’s ability to elicit an immune response, offering protection not only against DEV itself but also against other pathogens, such as the avian influenza (AI) virus.

Recombinant viral vectors are considered promising platforms for vaccines because they can express foreign antigens and stimulate both cellular and humoral immune responses without the need for external adjuvants. These vaccines are made up of viral particles that have been genetically altered to include one or more foreign genes that code for the desired antigens [1].

Our previous research focuses on developing a DEV-vectored vaccine targeting the H5 subtype of the AI virus. Given that DEV does not naturally infect chickens and, thus, no preexisting immunity to DEV is found in these animals, this vaccine could serve a dual purpose: protecting ducks from both DEV and AI infections while also mitigating the spread of AI when inoculated into chickens. The results from our DEV-H5 vector vaccine [8] indicate that it is safe and effective in ducks, successfully inducing protective immunity after viral challenge.

However, it is crucial to note that our vaccine demonstrated pathogenic effects in one-week-old chickens [8], stressing the need for careful consideration of the safety to non-target species in vaccine development. This highlights the importance of ongoing research to refine the vaccine formulation and assess its safety and efficacy in various avian species. Ultimately, our goal is to create a strong vaccine platform that can enhance poultry health and biosecurity, addressing pressing concerns in avian disease management.

The aim of this study was to determine tissue affinity and potential horizontal transmission of a developed DEV-H5 vector vaccine in 1-day-old chickens for an overall understanding of its pathogenicity. The data presented here provide evaluation parameters for further genomic modifications of DEV-vectored vaccines that are required to achieve complete safety and high immunogenicity in 1-day-old chickens.

## 2. Materials and Methods

### 2.1. Virus Strains and Cells

DEV-H5 and a commercial HVT-VP2 vector vaccine strain, containing, respectively, an AI H5 and an IBDV VP2 expression cassette, were used in the study [8,9] Figure 1.

Viruses were propagated in primary chicken embryo fibroblast cells (CEFs) derived from 11-day-old embryonated chicken eggs and maintained in Dulbecco’s Modified Eagle Medium (DMEM) supplemented with 10% fetal bovine serum and antibiotics.

### 2.2. Animals and Experimental Design

Fertile eggs from SPF white Leghorn chickens (Nanjing Zhushun Biotechnology Co., Nanjing, China) were incubated and hatched in our facilities. Chicks were kept in an isolation unit at the Jiangsu Academy of Agricultural Sciences (JAAS) throughout the experiment. All studies described here were approved by the Research Ethics Committee and Institutional Animal Care and Use Committee of Nanjing Agricultural University and JAAS (SYXK (Su) 2015-0019).

Ninety 1-day-old SPF chickens were randomly divided into three groups of thirty animals each. The groups were inoculated subcutaneously with 10^4^ TCID_50_ of DEV-H5, 10^4^ TCID_50_ of HVT-VP2, or DMEM (control group) in a 200 µL volume. To check for potential direct or indirect transmission, five additional animals were mixed with each of the vaccinated groups from day zero of the experiment. Animals were observed daily for clinical signs and mortality. At 3, 5, and 7 days post-infection (p.i.), three animals from each group were euthanized, and selected tissue samples and cloacal swabs were taken for the evaluation of viral loading and virus shedding and observation of gross/microscopic lesions. Environmental samples consisted of 0.5 g of bedding dissolved in 1 mL of PBS and collected from three evenly spaced locations. Gene expression of inflammatory cytokines and tight junction proteins at the intestine level was also assessed. At day 28 p.i., serum samples were taken for evaluation of antibody titers.

### 2.3. Viral DNA Extraction and qPCR

Fifty to one hundred mg of tissue or 200 μL of centrifuged liquid samples was extracted using a commercial DNA extraction kit according to the manufacturer’s instructions (Viral DNA/RNA extraction kit, Compurify Changzhou Biotech, Changzhou, China). Each tissue sample was precisely weighed and used later to calculate the DNA concentration per milligram. A NanoDrop spectrophotometer (Thermo Scientific, Verona, WI, USA) was used to assess DNA quality and quantity after extraction. Primers specific to HVT (Fw: 5′-CCCCCCTTCTTCGAGAGCC-3′ and Rv: 5′-AATATGACCATGTCCCCGGTG-3′) and DEV [10] DNAs were used in the qPCR assay. A qPCR reaction mix was prepared using a commercial qPCR kit (2xTsingke master qPCR mix with SYBR green, Tsingke Biotech, Nanjing, China) and consisted of 10 μL of SYBR Green Master Mix, 1 μL each of forward and reverse primer (10 μM), 2 μL of DNA template, and nuclease-free water to a final volume of 20 μL. Extracted samples were run in duplicate with the following cycling conditions: initial denaturation at 95 °C for 1 min, followed by 40 cycles of denaturation at 95 °C for 10 s, annealing at 60 °C for 30 s, and extension at 72 °C for 30 s. Viral load quantification in samples was based on a standard curve constructed of a 10-fold dilution series of a PCR product containing the target sequence with a known concentration.

### 2.4. RNA Extraction, cDNA Synthesis, and Gene Relative Expression

Total RNA was extracted from intestine samples using Trizol (Invitrogen, Carlsbad, CA, USA) according to the manufacturer’s instructions. Briefly, 50–100 mg of tissue samples was homogenized in Trizol reagent using a tissue homogenizer. Chloroform was added to the homogenate, and samples were centrifuged at 12,000 rpm for 15 min at 4 °C. The aqueous phase containing RNA was collected and mixed with isopropanol to precipitate the RNA. The RNA pellet was washed with 70% ethanol and resuspended in nuclease-free water. The concentration and purity of the RNA were determined using a NanoDrop spectrophotometer. First-strand cDNA was synthesized from extracted RNA using the High-Capacity cDNA Reverse Transcription Kit (Applied Biosystems, Foster, CA, USA). Briefly, 1 μg of total RNA was reverse transcribed into cDNA using random hexamers and the reverse transcriptase enzyme. The cDNA synthesis reaction was carried out at 42 °C for 2 min, followed by 37 °C for 15 min and 85 °C for 5 s. Quantitative real-time PCR (qPCR) was performed using SYBR Green Master Mix (Applied Biosystems) on a StepOnePlus Real-Time PCR System (Applied Biosystems). Primer sequences of genes for zonula occludens (ZO-1), occludin, interferon-α (IFN-α), and glyceraldehyde-3-phosphate dehydrogenase (GAPDH) were taken from Kong et al. [10]. The qPCR reaction mixture contained 10 μL of SYBR Green Master Mix, 1 μL each of forward and reverse primer (10 μM), 2 μL of cDNA template, and nuclease-free water to a final volume of 20 μL. The qPCR cycling conditions were as follows: 50 °C for 2 min, 95 °C for 2 min, followed by 40 cycles of 95 °C for 15 s and 60 °C for 1 min. The threshold cycle (Ct) values were determined for each sample and normalized to GAPDH using the 2^−ΔΔCt^ method [11]

### 2.5. Histopathology

Collected tissues at necropsy were fixed in 4% and sent to Wuhan Saiwei Biotechnology Co., Wuhan, China, for processing. Samples were embedded in paraffin, sectioned, and stained with hematoxylin-eosin.

### 2.6. Serological Tests

A hemagglutination inhibition assay was performed to determine the immunogenicity of the DEV-H5 virus. Briefly, serial two-fold dilutions of chicken serum samples were performed in duplicate in 96-well U-bottom plates, followed by the addition of 4 hemagglutination units of avian influenza [8]. Plates were incubated at room temperature for 30 min, and 0.5% chicken red blood cells were added to the virus/serum mixture and incubated at room temperature for another 30 min. The hemagglutination inhibition (HI) antibody titer was determined as the reciprocal of the highest dilution that completely prevented RBCs from agglutination. The Agar Gel Immunodiffusion Test was basically performed for the detection of antibody HVT-VP2 as described in [12].

### 2.7. Statistical Analysis

All data were subjected to the Student’s *t*-test with GraphPad Prism 9.0. Results are presented as mean ± standard deviation (SD), ns: not significant (*p* > 0.05); * *p* < 0.05, ** *p* < 0.01, *** *p* < 0.001.

## 3. Results

### 3.1. Morbidity and Mortality in 1-Day-Old Chickens

Starting at day 2 p.i., all DEV-H5 vaccinated birds showed clinical signs of duck plague, such as diarrhea, lethargy, loss of appetite, and inability to stand. A total of two, four, three, four, three, and three birds died at day 2, 3, 4, 5, 6, and 7 p.i., respectively, while the remaining ones were subjected to necropsy.

All HVT-VP2-vaccinated birds remained clinically healthy during the observation period. The weight gain trend of the chicks inoculated with HVT-VP2 was consistent with that of the control group, showing a steadily increasing trend, while the weight of the chicks inoculated with DEV-H5 decreased until the time of necropsy or death due to severity of infection (Figure 2).

### 3.2. Virus Replication and Shedding

To assess the replication and shedding of DEV-H5 and HVT-VP2 in inoculated animals, viral loads were quantified in the kidney, heart, intestine, liver, and spleen. On days 5 and 7 post-inoculation (p.i.), an increase in viral DNA copies per µL was detected in the kidney, spleen, intestine, and liver, whereas a decrease was observed in the heart on day 7 (Figure 3). Cloacal swabs and litter samples were collected to evaluate viral shedding. Only the chickens inoculated with DEV-H5 showed evidence of viral shedding, with the peak viral load recorded at day 5 p.i., reaching 4 log10 copies/µL (Figure 3). Furthermore, viral DNA was identified in bedding samples from this group, with maximum viral load titers of 1.9 log10 copies/µL also occurring at day 5 p.i. (Figure 4). In contrast, for animals inoculated with HVT-VP2, the spleen, bursa, and liver tissues were examined for virus replication. Viral loads began to rise from day 3 p.i., peaking on day 5 p.i. with titers ranging from 2.25 to 2.55 log10 copies/µL (Figure 5).

### 3.3. Gross and Microscopic Lesions

The next step involved investigating whether the DEV-H5 vector vaccination resulted in gross and microscopic lesions in the liver, lung, spleen, kidney, intestine, bursa, and heart, all of which are known to be susceptible to viral replication in ducks. By day 7 post-inoculation (p.i.), chickens inoculated with DEV-H5 exhibited notable abnormalities, including hemorrhage, congestion, necrosis, atrophy, and edema across all examined tissues (Figure 6). Microscopically, there was clear evidence of inflammatory cell infiltration and edema. Furthermore, significant manifestations of viral replication included extensive rupture of intestinal villi, inflammatory lesions in the lung, and an increased separation between cardiac muscle cells. By day 5 p.i., while the described lesions were less pronounced, they were still observable in the intestine, liver, kidney, lung, and heart. At the early stage of viral replication (day 3 p.i.), microscopic lesions were detected in the intestinal tissue of DEV-H5-inoculated chickens. In contrast, tissues selected from both the control and HVT-VP2 groups exhibited neither gross nor microscopic lesions.

### 3.4. Gene Expression Associated with Intestinal Barrier Function and Inflammatory Factors

When we evaluated the associated gene expression of DEV-H5 infection in intestinal tissue, the expression of IFN-α was statistically upregulated. The analysis of tight junction-associated proteins showed that DEV-H5 replication increased ZO-1 and occludin expression at 3 and 5 days p.i., followed by a significant increase at day 7 p.i. (Figure 7).

### 3.5. Serology

To assess in-contact transmission, non-infected chickens were cohoused with DEV-H5- and HVT-VP2-infected animals and evaluated for the presence of antibodies by HI and AGID, respectively. None of the in-contact animals developed antibodies against the AI virus or IBDV by the end of the experiment.

## 4. Discussion

This study aimed to determine tissue affinity and the potential horizontal transmission of a developed DEV-H5 vector vaccine in 1-day-old chickens for an overall understanding of its pathogenicity.

We have been working on the generation of a DEV-H5 vector vaccine candidate containing a synthesized HA expression cassette between the ORF of UL55 and LORF11 [8]. Animal studies showed that our DEV-H5 vector vaccine was safe in ducks and that one dose of the vaccine provided 100% protection against duck plague. Unexpectedly, while high levels of detectable anti-HA antibodies were induced, one dose of the vaccine was pathogenic when inoculated into three-week-old chickens [8]. Given this background, we aim to systematically evaluate the replication dynamics of our vector vaccine candidate in 1-day-old chickens. We decided to simultaneously assess the dynamics of a widely used HVT vector vaccine. HVT-VP2 has been proven to be a very convenient and safe vector in the development of recombinant vaccines [1].

After vaccination with our candidate DEV-H5 vector vaccine, birds showed signs of ataxia, severe enteritis, and watery diarrhea from day 2 p.i., and as expected, chickens lost body weight compared to the control and HVT-VP2 groups. Animals were found dead from day 2 p.i. onwards. At necropsy, gross and microscopical lesions were evident. Viral replication was detected in all investigated tissues. The result is consistent with that reported by Kong et al. [13], in which 1-day-old chickens were vaccinated with a DEV-H5 vector vaccine, and subsequently, high viral titers and pathogenic damage were present. Although DEV-H5 was excreted in feces and viral DNA was detected in litter samples, there were neither signs of infection nor antibody seroconversion in in-contact animals. Whether low shedding levels of DEV-H5 caused the lack of infection or whether chickens can only get infected through parenteral injection is yet to be investigated.

As expected, IFN-α was upregulated in response to DEV-H5 replication in epithelial cells of the intestine. IFN-α is a member of the interferon cytokine family that is induced during viral infection, mediating antiviral activities in the host [9,14]. Consistent with other studies [9], the expression of the tight junction proteins occludin and ZO-1 increased as the response to early acute infection. Tissue junctions are highly specialized membrane domains that have a strong barrier function, playing a critical role in preventing the spread and dissemination of viral pathogens [9,13,15]. Disruption of tight junctions of mucosal epithelia facilitates paracellular viral penetration and initiates systemic disease [16,17].

Various commercially available HVT-vectored vaccines are currently used to protect chicken flocks simultaneously against Marek’s disease and other important pathogens, such as AIV, IBDV, ILTV, and NDV [18]. HVT vector vaccines are safe to use and do not spread horizontally [18,19]. Consistently, here, 1-day-old birds inoculated with an HVT-VP2 vector vaccine neither developed clinical signs of infection nor gross/microscopic lesions. Viral replication was detected in the liver, spleen, and bursa, which are target lymphoid organs for HVT replication [20,21], while shedding was not detectable. In line with this, safety studies using the HVT-VP2 vector vaccine did not induce visible gross pathological changes or significant microscopical lesions of the bursa [19,20] and were not shed in the environment. However, it is important to mention the shortcomings associated with this control. These two vectors carry different foreign inserts, which could potentially influence the outcomes. In the future, comparative studies will be conducted using viral vectors carrying the same foreign antigen.

In conclusion, our work shows that our proposed DEV-H5 vector vaccine is not safe, has a broad tissue tropism, and does not spread horizontally when inoculated in 1-day-old chickens. The investigation provides valuable information for comparative assessment of the safety and tissue distribution of newly generated DEV-H5 vector vaccines. Further attenuation of our DEV-H5 vector vaccine through the deletion of immunomodulatory genes, such as glycoprotein E, glycoprotein I, and thymidine kinase, is currently being evaluated.

## Figures and Tables

**Figure 1 viruses-16-01681-f001:**
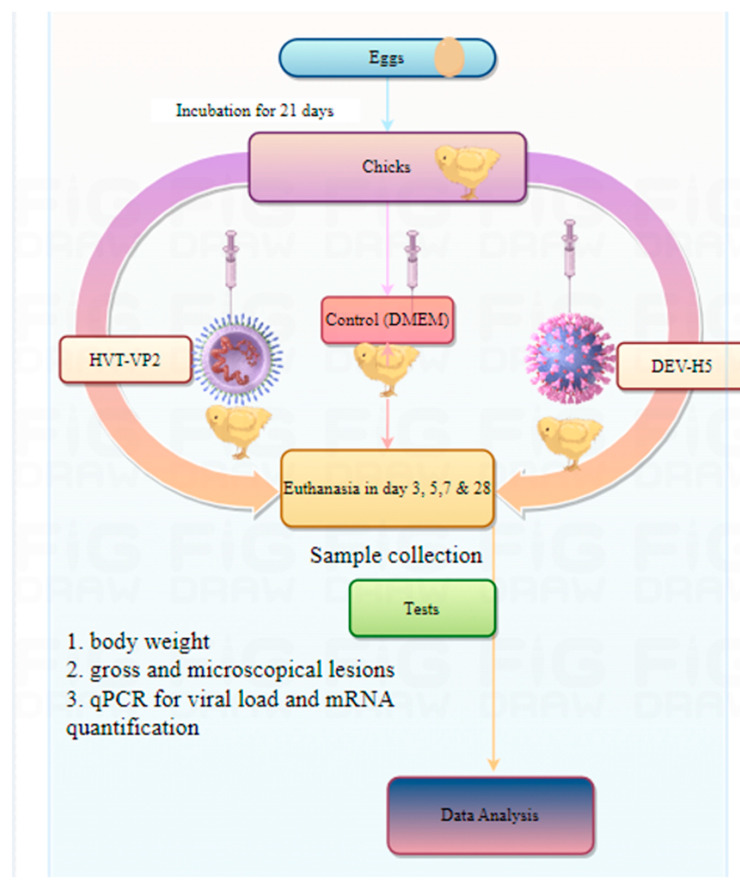
Graphic depiction of the experimental design.

**Figure 2 viruses-16-01681-f002:**
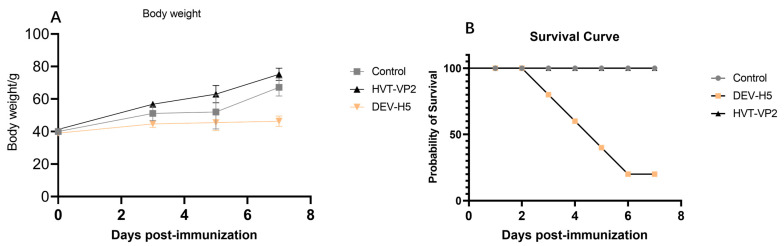
(**A**) Changes in body weight over the experimental period illustrating the average weight of chickens in the control, DEV-H5, and HVT-VP2 groups. (**B**) Survival rate of chickens for control, DEV-H5, and HVT-VP2 groups showing the proportion of surviving individuals. Note that the control and HVT-VP2 group curves overlap during the observation period.

**Figure 3 viruses-16-01681-f003:**
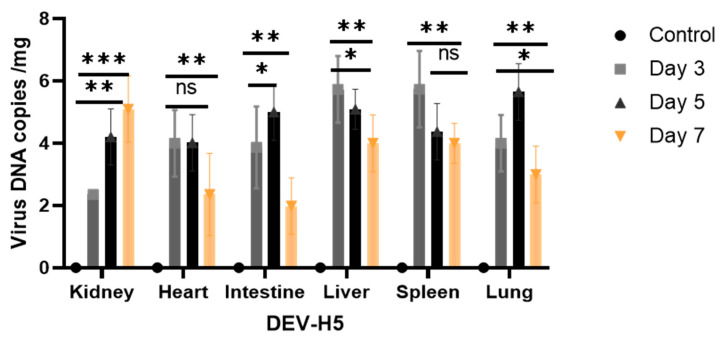
Quantification of viral DNA copy number in various tissue samples. The graph displays the mean viral DNA copy numbers (in copies per mg) detected in the kidney, heart, intestine, liver, and spleen samples. Statistical significance was determined using Graph Pad Prism version 9, with asterisks indicating significant differences between groups: * *p* < 0.05, ** *p* < 0.001 and, *** *p* < 0.0001), ns: not significant.

**Figure 4 viruses-16-01681-f004:**
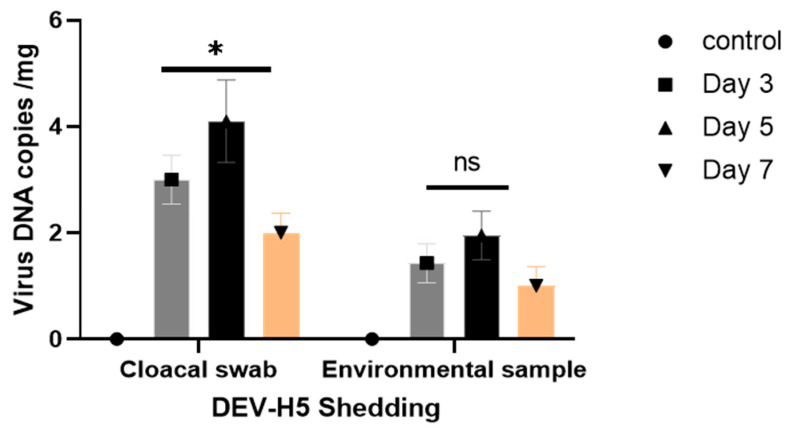
Quantification of viral DNA copy number in cloacal and environmental samples. The graph displays the mean viral DNA copy numbers (in copies per mg) detected in described samples. Statistical significance was determined using Graph Pad Prism version 9, with asterisks indicating significant differences between groups: * *p* < 0.05, ns: not significant.

**Figure 5 viruses-16-01681-f005:**
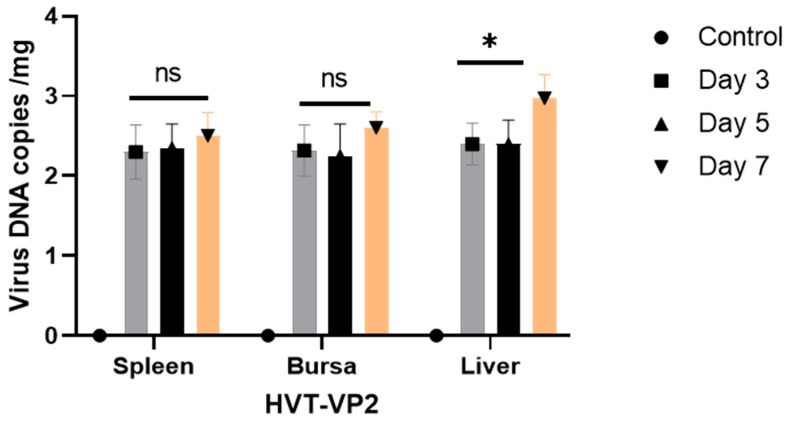
qPCR quantification of HVT-VP2 DNA copies in the spleen, bursa of Fabricius, and liver at days 3, 5, and 7 post-vaccination. Statistical analysis: * *p* < 0.05, ns: not significant.

**Figure 6 viruses-16-01681-f006:**
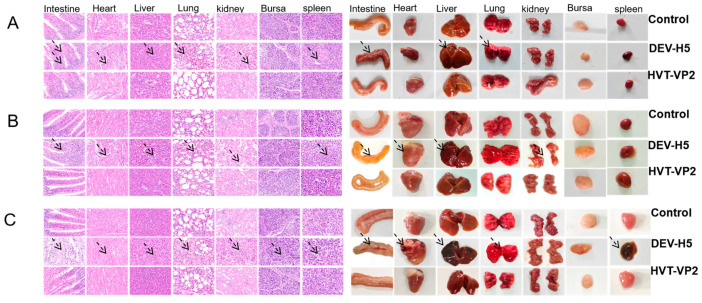
Gross and histopathological examination of tissue samples collected at 3 (**A**), 5 (**B**), and 7 (**C**) days p.i. with DEV-H5, HVT-VP2, and control groups. The control and HVT-VP2 groups exhibited no pathological changes throughout the study period. In contrast, the DEV-H5 group displayed significant pathological changes, as shown by arrows, at all assessed time points (days 3, 5, and 7), indicating the effects of the virus on the tissues.

**Figure 7 viruses-16-01681-f007:**
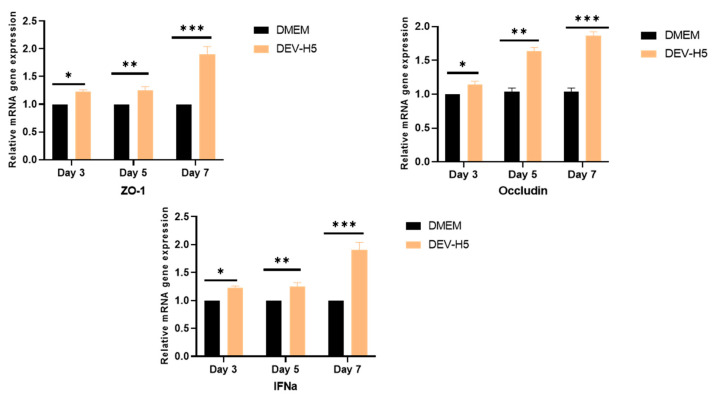
Assessment of mRNA levels for tight junction proteins in the intestine. The expression levels of ZO-1, IFN-α, and occludin were quantified to evaluate tight junction integrity. Data are presented as relative mRNA expression, normalized to housekeeping gene levels, and expressed as mean ± SEM. Significant differences between groups are indicated: * *p* < 0.05, ** *p* <0.001, *** *p* < 0.0001.

## Data Availability

All the relevant data are provided in this pape.

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
