# Peer review of "Evaluation of Tissue Tropism and Horizontal Transmission of a Duck Enteritis Virus Vectored Vaccine in One-Day-Old Chicken"

_viruses, 2024, doi:10.3390/v16111681_

Round 1
Reviewer 1 Report
Comments and Suggestions for Authors
The here presented manuscript by Abdulrahim et al. examines the pathology caused by vaccination with a duck enteritis virus (DEV) based vaccine vector carrying an avian influenza derived H5 protein in 1-day old chickens in comparison to a commercially available herpesvirus of turkey (HVT) vaccine vector containing a VP2 protein from the infectious bursal disease virus. The authors demonstrate, that although the DEV vector is benign in its original host, the duck, it is lethal in newly hatched chicks and that vaccination with the vector can results in significant viral replication across tissues resulting in detectable histopathological changes and virus shedding. Although virus DNA in detectable in environmental samples, unvaccinated, DEV naïve chicks don’t get infected by their vaccinated cagemates, indicating that the vaccine vectors is not spread horizontally. The result of this study will enable the investigators to proceed with their vector development as they now have a readout, they can track to determined what introduced attenuation can result in reduced pathology and increased survival of their vaccinated animals.
While the here presented work does not require additional experiments, significant changes to the manuscript are required as some information is missing, and some results presented in this study should be discussed and put into context.
List of criticisms:
1) The title of this manuscript, “Tissue affinity and horizontal transmission of a duck enteritis 2 virus vectored vaccine in one-day-old chicken” shoud probably read something like “Evaluation of tissue tropism and horizontal transmission of a duck enteritis 2 virus vectored vaccine in one-day-old chicken”.
2) Line 25: “All inoculated chickens succumbed to infection …”.
As we are talking about a vaccine vector, these animals technically die after vaccination.
3) Line 30: “As control, birds inoculated with a commercial recombinant turkey herpesvirus expressing the VP2 protein of infectious bursal disease virus (HVT-VP2) vector vaccine showed neither clinical signs nor mortality.”
While not a bad control, this control is imperfect as the two vectors carry different vaccine inserts that could technically be responsible for the observed pathology and signs of disease or the lack thereof. This needs to be discussed on the manuscript.
4) Line 46: ”Duck enteritis virus (DEV), an alpha-herpesvirus, carriages a significant threat to water fowl, particularly ducks, geese, and swans, with mortality rates reaching as 100% in infected ducks [6].”
The authors likely meant to say “carries” instead of “carriages”.
5) Line 53: “Given that chickens do not have immunity to DEV, …”
It might be worth mentioning that DEV does not naturally infect chickens and that this is the reason why there are not preexisting immune responses in these animals.
6) Line 55: “The results from our DEV-H5 vector vaccine [8] indicate that it is safe and effective in ducks, successfully inducing protective immunity after viral challenge.”
The authors should give more information about the vaccine vectors. For instance, Where were the transgenes introduced? what other modifications were made to the vector backbone? How was the virus given and what dose was used? Also, what does effective mean in this context? Did the authors previously perform a challenge study and vaccination with the vectors provided protective immunity?
7) Line 58: “However, it is crucial to note that our vaccine demonstrated pathogenic effects in one-week-old chickens, …”
Please see the comment above. When these experiments were performed, what exact vector was used? What modifications were made to the vector and how was the vector given? What pathological changes were observed?
8) Line 60: “This highlights the importance of ongoing research to refine the vaccine formulation and assess its safety and efficacy in various avian species.”
Judging by the data the authors are presented here, it looks like the vaccine formulation was probably fine, but the vector backbone itself was likely not attenuated enough.
9) Line 71: “DEV-H5 and a commercial HVT-VP2 vector vaccine strains, containing respectively an AI H5 and an IBDV VP2 expression cassettes were used in the study [[8],[9]].”
This might be a good place to give a detailed description of the vaccine vectors and maybe include a schematic to allow the reader to understand what exactly was used.
10) Line 96: “Each tissue sample was precisely weighed and used later to calculate the DNA concentration per microliter.”
I am not too sure I understand how a weight can help to express a DNA concentration per microliter, which is a unit of volume?
11) Figure 1.
It is probably more common to express the weight gain or loss as changes to the original weight set to 100%, but the current presentation is acceptable. In figure B, how many animals survived? This information was not provided, and these results were not discussed in the main text. Did all test animals perish? Or did some survive and did these animals make a full recovery?
12) Line 166: “… viral loads were quantified in the kidney, heart, intestine, liver, and spleen…“
Why were these specific tissues chosen?
13) Figures 2, 3 and 4:
To normalize all these samples to each other, the viral genome copy numbers should be expressed a copies per ug DNA, not copies per microliter. The volume can change by diluting or concentration the sample.
14) Line 194: “The next step involved investigating whether the DEV-H5 vector vaccination resulted in gross and microscopic lesions in the liver, lung, spleen, kidney, intestine, bursa, and heart, all of which are known to be susceptible to viral replication.”
Is the statement that the authors are making here based on prior results in chicken infected with DEV, or is that statement an extrapolation of results resulted generated in ducks?
15) Figure 5
The individual pictures presented in this figure is very small and the mentioned lesions are extremely hard to see. The authors should make this figure substantially larger and include arrows to point out what histopathological changes they are referring in the manuscript to allow for easier comprehension and interpretation of the data.
16) Line 214: “When we evaluated the associated gene expression of DEV-H5 infection in intestinal tissue, the expression of IFN-α was statistically up-regulated. Analysis of tight junction-associated proteins showed that DEV-H5 replication increased ZO-1 and occludin expression at 3 and 5 days p.i. followed by a significant increase at day 7 p.i.”
What was the scientific rational that the authors are focusing on the intestines? Why did the tight junctions represent in valuable target for investigation? What were the reason the authors decided to study the particular genes investigated in this manuscript? What does the increased gene expression mean for the virus and the vector system? Do these results indicate that there might be a difference in protein expression? How would the virus benefit from the here observed phenotype? Is this phenotype also observed in ducks infected with DEV?
17) Line 224: “None of the in-contact animals developed antibodies against AI virus or IBDV by the end of the experiment.”
Has it been established that the wild type virus, not the genetically modified vector can be transmitted from between immunocompetent chickens? Or is s.q. vaccination required for infection?
18) Line 233: “We have been working in the generation of a DEV-H5 vector vaccine candidate containing a synthesized HA expression cassette between the ORF of UL55 and LORF11[8].”
As mentioned above, can the authors provide a schematic showing what was alterations were made to the vector backbone, where the transgene is expressed and what promoter is driving transgene expression? Have equivalent changes been made to the commercial HVT vector?
19) Line 251: “The finding suggests that the low-level shedding of DEV-H5-vaccinated chickens restricted propagation to susceptible animals.”
Not necessarily, could it be possible that the virus cannot infect chickens unless it is injected into the animal? Has it been established that chicken can be naturally infected like ducks with this virus?

The level of English used by the authors is fine. The manuscript includes some mistakes, but they do not affect the understandability of the presented work.
Author Response
Answer to reviewer #1
We would like to take this opportunity to express our sincere gratitude for your important feedback and insights regarding our manuscript “Tissue Affinity and Horizontal Transmission of a Duck Enteritis Virus Vectored Vaccine in One-Day-Old Chickens”. We take your concerns seriously and have addressed them to the best of our abilities. We believe that your comments have significantly improved the quality of our manuscript. We revised and corrected our manuscript as follows:
- The title of this manuscript, “Tissue affinity and horizontal transmission of a duck enteritis 2 virus vectored vaccine in one-day-old chicken” should probably read something like “Evaluation of tissue tropism and horizontal transmission of a duck enteritis 2 virus vectored vaccine in one-day-old chicken”
Response: As you suggested, we have changed the title of this work accordingly.
- Line 25: “All inoculated chickens succumbed to infection …”.
As we are talking about a vaccine vector, these animals technically die after vaccination.
Response: We agree with your comment and changed succumbed to the more appropriate died (lanes 25 to 27 and lanes 181-183).
- Line 30: “As control, birds inoculated with a commercial recombinant turkey herpesvirus
expressing the VP2 protein of infectious bursal disease virus (HVT-VP2) vector vaccine
showed neither clinical signs nor mortality.
While not a bad control, this control is imperfect as the two vectors carry different vaccine inserts that could technically be responsible for the observed pathology and signs of disease or the lack thereof. This needs to be discussed on the manuscript.
Response: Thanks for your comment. We discussed it consequently (see discussion, highlighted, lanes 300-304).
- Line 46: “Duck enteritis virus (DEV), an alpha-herpesvirus, carriages a significant threat
to water fowl, particularly ducks, geese, and swans, with mortality rates reaching as 100% in infected ducks [6].”
The authors likely meant to say “carries” instead of “carriages”.
Response: Thanks for reading the manuscript thoroughly. We amended the spelling error (lane 46).
- Line 53: “Given that chickens do not have immunity to DEV, …”
It might be worth mentioning that DEV does not naturally infect chickens and that this is the reason why there are not preexisting immune responses in these animals.
Response: Appreciate your input. We revised the sentence and we subsequently modified it (lanes 58 to 61)
- Line 55: “The results from our DEV-H5 vector vaccine [8] indicate that it is safe and
effective in ducks, successfully inducing protective immunity after viral challenge.”
The authors should give more information about the vaccine vectors. For instance, where were the transgenes introduced? what other modifications were made to the vector backbone? How was the virus given and what dose was used? Also, what does effective mean in this context? Did the authors previously perform a challenge study and vaccination with the vectors provided protective immunity.
Response: Appreciate your feedback. Our previous work is appropriately referenced and readers can find more details.
- Line 58: “However, it is crucial to note that our vaccine demonstrated pathogenic effects in one-week-old chickens, …”
Please see the comment above. When these experiments were performed, what exact
vector was used? What modifications were made to the vector and how was the vector given? What pathological changes were observed?
Response: Detailed information is provided in our referenced manuscript.
- Line 60: “This highlights the importance of ongoing research to refine the vaccine
formulation and assess its safety and efficacy in various avian species.”
Judging by the data the authors are presented here, it looks like the vaccine formulation was probably fine, but the vector backbone itself was likely not attenuated enough
Response: Thank you for your opinion. That is why we mention in the discussion section that a gene deletion is needed to improve vector vaccine safety (lanes 308-306).
- Line 71: “DEV-H5 and a commercial HVT-VP2 vector vaccine strains, containing respectively an AI H5 and an IBDV VP2 expression cassettes were used in the study
[[8],[9]].”
This might be a good place to give a detailed description of the vaccine vectors and maybe include a schematic to allow the reader to understand what exactly was used.
Response: Thank you for your comment. We added the information to the manuscript (introduction part, lanes 52 to 56).
- Line 96: “Each tissue sample was precisely weighed and used later to calculate the DNA
concentration per microliter.”
I am not too sure I understand how a weight can help to express a DNA concentration per microliter, which is a unit of volume?
Response: Thanks for your valuable suggestion. We agree with your comment. Our procedure consisted in extracting DNA from up to 100 mg of tissues using a column-based method, used the same amount of elution buffer with 2 ul of extracted DNA used in the reaction. As the tissues samples were precisely weighed before the extraction, viral loads were normalized to 1 mg of tissue (virus DNA copies/mg)
- Figure 1.
It is probably more common to express the weight gain or loss as changes to the original
weight set to 100%, but the current presentation is acceptable. In figure B, how many
animals survived? This information was not provided, and these results were not discussed in the main text. Did all test animals perish? Or did some survive and did these animals make a full recovery?
Response: We have added the information and discussed in the text (lanes 25-27, lanes 182-183)
- Line 166: “… viral loads were quantified in the kidney, heart, intestine, liver, and spleen…“
Why were these specific tissues chosen?
Response: DEV has a broad tissue tropism in ducks, which has experimentally been demonstrated. We subsequently selected the organs based on previous works in ducks. The information was included in lane 225.
- Figures 2, 3 and 4:
To normalize all these samples to each other, the viral genome copy numbers should be
expressed a copies per ug DNA, not copies per microliter. The volume can change by
diluting or concentration the sample.
Response: As we mentioned above (point 10) virus DNA copies are now expressed to mg of tissue through the paper.
- Line 194: “The next step involved investigating whether the DEV-H5 vector vaccination
resulted in gross and microscopic lesions in the liver, lung, spleen, kidney, intestine, bursa, and heart, all of which are known to be susceptible to viral replication.”
Is the statement that the authors are making here based on prior results in chicken infected with DEV, or is that statement an extrapolation of results resulted generated in ducks?
Response: Thanks for your observation. The statement is based according to tissue tropism in ducks. Consequently, we inform the readers by adding “in ducks” at the end of the sentence (lane 225)
- Figure 5
The individual pictures presented in this figure is very small and the mentioned lesions are
extremely hard to see. The authors should make this figure substantially larger and include
arrows to point out what histopathological changes they are referring in the manuscript to allow for easier comprehension and interpretation of the data.
Response: Thanks for your observation. Figure size has been changed and arrows are included.
- Line 214: “When we evaluated the associated gene expression of DEV-H5 infection in intestinal tissue, the expression of IFN-α was statistically up-regulated. Analysis of tight junction-associated proteins showed that DEV-H5 replication increased ZO-1 and occludin expression at 3 and 5 days p.i. followed by a significant increase at day 7 p.i. ”
What was the scientific rational that the authors are focusing on the intestines? Why did the tight junctions represent in valuable target for investigation? What were the reason the authors decided to study the genes investigated in this manuscript? What does the increased gene expression mean for the virus and the vector system? Do these results indicate that there might be a difference in protein expression? How would the virus benefit from the here observed phenotype? Is this phenotype also observed in ducks infected with DEV?
Response: In a susceptible host, DEV replicates primarily in the mucosa of the digestive tract, and then spreads to other organs. The mentioned cytokine and proteins expressions are affected due to viral replication. However, we do not have the information regarding our DEV-H5 vectored vaccine. We consequently investigated the expressions of the above-mentioned parameters in this work.
- Line 224: “None of the in-contact animals developed antibodies against AI virus or IBDV by the end of the experiment.”
Has it been established that the wild type virus, not the genetically modified vector can be transmitted from between immunocompetent chickens? Or is s.q. vaccination required for infection?
Response: Previous research addressed this issue (Front. Microbiol. 13:979368. doi: 10.3389/fmicb.2022.979368). BUT, to our knowledge no report about vectored vaccine up to now.
- Line 233: “We have been working in the generation of a DEV-H5 vector vaccine candidate containing a synthesized HA expression cassette between the ORF of UL55 and LORF11[8].”
As mentioned above, ca authors provide a schematic showing what was alterations
were made to the vector backbone, where the transgene is expressed and what promoter
is driving transgene expression? Have equivalent changes been made to the commercial HVT vector?
Response: Thanks for pointing this out. The issue is appropriately discussed in our previous work which is referenced through the manuscript.
- Line 251: “The finding suggests that the low-level shedding of DEV-H5-vaccinated chickens restricted propagation to susceptible animals.”
Not necessarily, could it be possible that the virus cannot infect chickens unless it is injected into the animal? Has it been established that chicken can be naturally infected like ducks with this virus?
Response: Thanks for this valuable comment. We have then appropriately commented this fact in the discussion section (lanes 281-282).

Reviewer 2 Report
Comments and Suggestions for Authors
In this study, the authors describe the use of the duck enteritis virus as a vector vaccine expressing the hemagglutinin of HPAIV H5 (DEV-H5) in neonate chickens. The authors reported a high lethality and remarkable pathology in the animals inoculated with the construct in contrast to the safety of the commercial vectored vaccine HVT-VP2. This reviewer congratulates the authors for their interest in publishing their findings. Although they can be considered negative results, they are also results after all and therefore, important to the progress of this science. However, some issues need to be considered to improve the quality of the paper.
The title does not reflect the results of the work, it is descriptive and general, but inaccurate. It can be improved accordingly.
Lines 58-59: Please add the reference from the previous results of vaccine pathogenicity, otherwise state “unpublished results”.
A scheme of the study may improve the quality of the paper, it can be included in the main body of the manuscript or the supplementary data.
Lines 84-85: It is unclear when (day, timepoint) the five additional naïve animals to check virus transmission were included in the study. Please add this information.
The histopathology section is poorly described. Please at least specify briefly the procedures carried out with the sampled tissues.
Fig 1B. The survival curve is difficult to perceive, two groups are overlapped. It would be more convenient to draw it as a survival graph in “stair” form.
Viral loads from DEV-H5 were represented extensively for different organs, cloacal swabs and bedding samples, but not for the control vaccine HVT-VP2. To make it comparable, it would be necessary to represent the same type of samples in both vector vaccines. For instance, the bursa is not depicted for the DEV-H5, and either kidney or cloacal swabs for HVT-VP2. Please add it if possible.
Figure 5. The histopathologic examination is difficult to observe. Please represent it in a larger format and highlight the infiltrations and lesions (with arrows) described in the caption and the text.
Section 3.4. Gene expression assessment. If the control group is DMEM-treated, please add it to the figures.
Serology results are also poorly described, please show the results in a table or a graph. Include the HI titres of the chickens of the study to compare their immune response. Please add the complete name of the strain used for HIA in the materials and methods section.
The authors relate there was no horizontal transmission when they measured the humoral response. But this transmission was not even mentioned in other sections of the results, for instance in the viral loads. Please clarify in the text whether the naïve animals included in the experiment became “infected” and the viral load of their organs. This may be a better measure of the transmissibility of the vector vaccine rather than only the humoral response “per se”.
In the discussion, it would be interesting to have insight into possible strategies that can be carried out to attenuate this vector and use it as a vaccine in the future.
Author Response
Answer to reviewer #2
Thank you very much for your kind suggestions and comments. We take your concerns seriously and have addressed them to the best of our abilities. We firmly believe that your comments and those of reviewer #1 have significantly improved this manuscript. We revised and corrected our manuscript as follows:
- The title does not reflect the results of the work, it is descriptive and general, but inaccurate. It can be improved accordingly.
Response: Thanks for your comment. We have changed the title of this work accordingly.
- Lines 58-59: Please add the reference from the previous results of vaccine pathogenicity, otherwise state “unpublished results”.
Response: Thanks for your observation. Reference has been added (lane 56).
- A scheme of the study may improve the quality of the paper, it can be included in the main body of the manuscript or the supplementary data.
Response: Thanks for your suggestion to improve the quality of the manuscript. A scheme of the study has been added to the paper (Figure 1).
- Lines 84-85: It is unclear when (day, timepoint) the five additional naïve animals to check virus transmission were included in the study. Please add this information.
Response: We now included that naïve animals were mixed with each of the vaccinated groups from day zero of the experiment (lane 92)
- The histopathology section is poorly described. Please at least specify briefly the procedures carried out with the sampled tissues.
Response: Thanks for your observation. We included additional information as suggested (lanes 160 to 161)
- Fig 1B. The survival curve is difficult to perceive, two groups are overlapped. It would be more convenient to draw it as a survival graph in “stair” form.
Response: Control and HVT-VP2 were 100% safe and they look overlapped in the figure. Following your suggestion, this fact is remarked in the figure legend.
- Viral loads from DEV-H5 were represented extensively for different organs, cloacal swabs and bedding samples, but not for the control vaccine HVT-VP2. To make it comparable, it would be necessary to represent the same type of samples in both vector vaccines. For instance, the bursa is not depicted for the DEV-H5, and either kidney or cloacal swabs for HVT-VP2. Please add it if possible.
Response: Thank you for suggestion. We did not want to overload our manuscript with figures or tables. As HVT-VP2 vector virus DNA was only detected in spleen, liver and bursa, we decided to include a figure containing these results.
- Figure 5. The histopathologic examination is difficult to observe. Please represent it in a larger format and highlight the infiltrations and lesions (with arrows) described in the caption and the text.
Response: We have followed your comments. The figure size is more appropriate and arrows were added.
- Section 3.4. Gene expression assessment. If the control group is DMEM-treated, please add it to the figures.
Response: Thanks for your observation. It has been appropriately added.
- Serology results are also poorly described, please show the results in a table or a graph. Include the HI titres of the chickens of the study to compare their immune response. Please add the complete name of the strain used for HIA in the materials and methods section.
Response: Thanks for your comment. Serological tests were applied to in-contact animals to address horizontal transmission. The presence of antibodies was not detected in this population; thus, we did not show it in a table. The complete name has been added as suggested (lane 166).
- The authors relate there was no horizontal transmission when they measured the humoral response. But this transmission was not even mentioned in other sections of the results, for instance in the viral loads. Please clarify in the text whether the naïve animals included in the experiment became “infected” and the viral load of their organs. This may be a better measure of the transmissibility of the vector vaccine rather than only the humoral response “per se”.
Response: Thanks for your observation. This point is presented in the results and discussion sections.
- In the discussion, it would be interesting to have insight into possible strategies that can be carried out to attenuate this vector and use it as a vaccine in the future.
Response: Thanks for your comment. Following your suggestion, we included strategies that we are evaluating for attenuation (last paragraph, highlighted).
